# The Role of Digital Technology in Curbing COVID-19

**DOI:** 10.3390/ijerph19148287

**Published:** 2022-07-07

**Authors:** Noha S. Alghamdi, Saeed M. Alghamdi

**Affiliations:** 1Business School, Imperial College London, London SW7 2BX, UK; noha.alghamdi20@imperial.ac.uk; 2National Heart and Lung Institution, Imperial College London, London SW3 6LY, UK; 3Respiratory Care Program, Clinical Technology Department, College of Applied Medical Science, Umm Al-Qura University, Makkah 24382, Saudi Arabia

**Keywords:** digital technology, digital health, COVID-19, health innovation, health informatics

## Abstract

Introduction: Using digital technology to provide support, medical consultations, healthcare services, and to track the spread of the coronavirus has been identified as an important solution to curb the transmission of the virus. This research paper aims to (1) summarize the digital technologies used during the COVID-19 pandemic to mitigate the transmission of the COVID-19; (2) establish the extent to which digital technology applications have facilitated mitigation of the spread of COVID-19; and (3) explore the facilitators and barriers that impact the usability of digital technologies throughout the pandemic. Methods: A rapid electronic search following Preferred Reporting Items for Systematic Reviews and Meta-Analyses (PRISMA) guidelines was conducted of available records up to June 2022 on the medical databases PubMed, Ovid, Embase, CINHAIL, the Cochrane Library, Web of Science, and Google Scholar. Results: An increasing number and variety of digital health applications have been available throughout the pandemic, such as telehealth, smartphone mobile health apps, machine learning, and artificial intelligence. Each technology has played a particular role in curbing COVID-19 transmission. Different users have gained benefits from using digital technology during the COVID-19 pandemic and different determinants have contributed to accelerating the wheel of digital technology implementation during the pandemic. Conclusion: Digital health during the COVID-19 pandemic has evolved very rapidly, with different applications and roles aimed at curbing the pandemic.

## 1. Introduction

In December 2019, the Corona virus disease 2019 (COVID-19) was discovered in Wuhan, China [1]. Then, in a very short time, the virus spread worldwide. To March 2022, records show that almost more than 520 million people have been infected with COVID-19 and more than 4 million people have died of the disease [2]. COVID-19 has been defined as an acute respiratory syndrome with a continuous cough, a high-grade fever, and shortness of breath [1]. Because of the rapid spread of COVID-19, the World Health Organization (WHO) declared it a pandemic [3]. Around the world, many healthcare systems have been affected and challenged due to the enormous number of cases that need healthcare services. 

In general, the COVID-19 pandemic has created many challenges for both individuals and healthcare systems. At the population level, public authorities and governments introduced lockdowns and quarantine to prevent the spread of the virus [4], since it was discovered that the virus is more active in social areas. The quarantine, stay-at-home, and social distancing policies were implemented to control the spread of the infection [4,5]. Research indicates that many healthcare systems were vulnerable in the face of such a pandemic, as it has increased existing challenges such as staff shortages, personal protective equipment (PPE) shortages, and intensive care unit (ICU) and hospital bed capacities [5,6,7]. This extraordinary situation imposes extra challenges on the existing healthcare systems. 

### Digital Health Solutions

Using digital technology to provide support, medical consultations, deliver healthcare services, and track the spread of the virus has been identified as a vital solution to help to curb the transmission of the virus [8]. 

Although digital technology in healthcare services was introduced decades ago as telehealth or remote healthcare services, the presence of the pandemic has dramatically boosted its applications and development as an important factor to mitigate the disease and to break the cycle of disease transmission. Digital health solutions have both hard and soft innovations. Hard health innovations involve the remote use of medical devices, and soft health innovations are defined as the knowledge to operate and monitor the devices, as well as the patients [9].

The terminology associated with the digital technology used to provide healthcare during the pandemic reflects the sources, mode, or the services that accompany the technology [10]. For example, ‘telehealth’ is the terminology used for telephone services or internet services that provide routine check-ups for confirmed COVID-19 patients, such as monitoring vital signs and tracking symptoms of the disease [10,11]. In another example, the electronic intensive care unit (eICU), the healthcare provider uses a live video camera to monitor ICU patients in another hospital, and to provide support for other healthcare providers [11]. A summary of the various digital health terminologies and their uses during COVID-19 is provided in the online Appendix A.

The pandemic has been a powerful trigger for accelerating the implementation and adoption of digital technologies to mitigate the spread of COVID-19 and to provide healthcare. This is confirmed in the available literature, which presents promising outcomes of using digital solutions to control the spread of the virus [5,8,12,13,14,15,16,17,18,19,20,21]. Other literature has focused on facilitators and barriers to the implementation, adoption, and sustainability of digital technology during COVID-19 and in future pandemics [10,22,23,24,25]. Accordingly, we can see that digital technology applications in healthcare delivery are a trending topic, and they may become the mainstream mode of care delivery. Even though healthcare services are improving because of digital health technology use, more understanding is needed. Therefore, the paper aims to summarize and synthesize the currently available evidence that assesses the use of digital technology to provide healthcare during the COVID-19 pandemic. Specifically, this research paper will answer the following research questions:Which digital technologies have been used during the COVID-19 pandemic to mitigate the transmission of the virus?To what extent have digital technology applications helped to mitigate the spread of COVID-19? Which facilitators and barriers have contributed to the usability of digital technologies during the COVID-19 pandemic? 

## 2. Methods

A rapid electronic search of available records up to June 2022 was carried out on medical databases to identify relevant articles. The keywords “Digital technology”, “Digital health”, “COVID-19”, “Health innovations”, “Health informatics”, “Health economy”, and “Health policy” were searched in the following databases: PubMed, Ovid, Embase, CINHAIL, the Cochrane Library, Web of Science, and Google Scholar. Studies examining, summarizing, or criticizing the use of digital technology in response to COVID-19 were included in the study. Non-English publications as well as articles that were not relevant to the outbreak of COVID-19 were excluded. The PRISMA guidelines were followed, when possible, in this systematic review. A standardized spreadsheet was used to collect data independently. Then, the authors gathered to review, discuss, and agree on the final findings. The final findings were reported in narrative form. We registered this review to Open Science Framework repository (Registration: https://doi.org/10.17605/OSF.IO/CFRHD). 

## 3. Results

### 3.1. Which Digital Technologies Have Been Used during COVID-19 to Mitigate the Transmission of the Virus?

Reducing the transmission of the virus was the first health priority in countries during the initial outbreak of COVID-19 [26]. A wide variety of digital technologies and healthcare innovations were initiated or activated to help in care delivery, to screen infected people or track them, to minimize the infection and mortality rates, to estimate the number of infections, and to flatten the curve [20]. 

Telehealth, or providing management over distance via available technology (telephone calls, mobile apps, and video calls), was the first line of defense against transmission of the virus. Telehealth helps to reinforce social distancing and enables patients to avoid exposure to large groups of people in the waiting areas of clinical centers [27]. This application is supported by the recommendations from the WHO and by governments implementing strict quarantine and lockdowns [28].

Artificial intelligence (AI) via infrared thermal cameras and thermometers has been installed in crowded places such as airports and other areas of public transportation, schools, and workplaces to identify and screen infected people and monitor their temperature, as a high temperature (fever) is one of the symptoms of the disease [19,29,30]. In addition, polymerase chain reaction (PCR) tests, other laboratory tests, and digital technologies such as mobile health apps that screen symptomatic and non-symptomatic COVID-19 patients have been implemented to gather real-time data about infected and non-infected people in several countries [31]. 

Mathematical models and machine learning (ML) have also enlightened researchers, health authorities, and healthcare systems about the expected number of infected people and/or mortality rates and have estimated the extent to which the infection will spread in particular locations [32]. The second and third waves of the pandemic have been identified before they happen from real-time data and from applying mathematical models and ML [33]. 

Contact, tracing, monitoring, and strict quarantining of patients have also been supported using digital technology and healthcare innovations. For example, in Saudi Arabia, the Minister of Health launched a mobile health application named Seha, Tawakkalna and Tabaud to allow contact tracing if an individual has a confirmed COVID-19 diagnosis [24]. The app uses distance points to notify an individual anonymously if they are close to a person who has a confirmed COVID-19 diagnosis [24,34]. Another example is the national health service (NHS) mobile application that enables people to scan the barcode before entering commercial and social areas [35,36,37]. This application provides information to the health services about the capacity of the location and/or if there are any infected cases [35]. Also, these mobile applications allow people to book their vaccination appointments and to provide proof that they have had completed their vaccinations [38].

### 3.2. Determinants of Digital Health during COVID-19 

Before planning to implement digital health solutions to mitigate COVID-19, the stakeholders or policymakers must consider the determinants that impact adaptations and adherence (Figure 1). One important determinant is ensuring easy access to basic and advanced communication technologies and/or digital health platforms for individuals. Another determinant is the existence of well-trained healthcare providers to lead and operate the platform, as well as a 24/7 technical support team to fix any technical issues. Third, the proposed digital health platforms must be secure for data transformations. Fourth, the platform users’ needs must be considered, for example, hearing and visual impairments in elderly people [39].

### 3.3. To What Extent Have Digital Technology Applications Helped to Mitigate the Spread of COVID-19? 

#### 3.3.1. Telehealth Has Helped to Follow Up and Treat People with Chronic Diseases

A systematic review reported that digital technology services have been the dominant strategy during the pandemic, especially for self-isolating people [40]. The systematic review included eight studies that demonstrate that using telehealth applications in medical fields has the potential to mitigate the spread of the virus [40] Telehealth is a significant way to maintain social distance and minimize the risk of virus transmission. A variety of telehealth techniques have been used, such as WhatsApp, social media platforms, and live video conferencing through Skype or Apple FaceTime. Furthermore, telehealth is useful in providing elective care and routine check-ups for people who do not need urgent care or invasive procedures [40]. However, social media platforms have contributed to propagating false information and rumors about vaccines and virus mutations. This systematic review was one of the first pieces of evidence about the benefits of digital health during the pandemic, but we think it lacks the quantitative data to evaluate the extent to which telehealth has helped to decrease the infection rate (Table 1). 

Another systematic review and meta-analysis reviewed the uptake level of the telehealth application in general for people with chronic respiratory disease. The systematic review reported the clinical trials that use telehealth applications to monitor or deliver care for those populations. The results showed that the acceptance rate was 80%, versus a low dropout rate of 19%. This was the normal response to the uptake of telehealth interventions before the pandemic occurred [47]. These results indicate that the adoption of telehealth, as well as awareness of the benefits of using telehealth, has reached an advanced level among people with chronic diseases. Recent cross-sectional data showed that the level of uptake and usability has been scaled up to 85% during the pandemic, and the use of telehealth was associated with a high satisfaction rate of 94%; patients were happy to attend telehealth sessions instead of in-person sessions [48]. However, these studies lack of qualitative data about the facilitators and barriers associated with digital health usability during the COVID-19. 

#### 3.3.2. Tele-Education to Support Professional Training 

The burden of the pandemic on medical education and training is obvious. There has been an interruption in direct management, training sessions, workshops, and medical education. Tele-education and/or remote training via ZOOM, Skype, or Microsoft Teams has replaced traditional methods in teaching and training programs, even in units such as ICUs [17]. This experience has conflicting results in the literature [49,50]. We believe that, in medical education, tele-education and tele-training experience could not be the same as in-person training for deep procedures and intensive interventions. 

#### 3.3.3. Virtual Intensive Care Unit 

This application of telehealth in the ICU has dramatically increased—due to the COVID-19 outbreak—in the last two years to overcome the shortage of healthcare professionals and to support family visits to critically ill patients [16]. Cross-sectional data about applying a virtual ICU during the pandemic showed positive results in decreasing physiological stress, anxiety, and improving staff morale [51]. Also, a virtual ICU is a very effective way to share feelings of happiness and relief between family members [16]. It was obvious that this approach facilitated experts’ shared opinions during the COVID-19 pandemic, as well as overcoming staff shortages. 

#### 3.3.4. Virtual Reality Helps in the Management of Pain and Physical Therapy

Virtual reality (VR) has also played a key role in handling cases during the pandemic. One of the most important uses for VR is in pain management for people who suffer from back pain or other chronic pain. The concept of using virtual reality to deliver care to those people has been proven by the experts as a feasible and powerful tool in healthcare delivery during the pandemic. Also, VR can provide a creative way to encourage physical activity among people enrolled in pulmonary rehabilitation programs [18]. Virtual reality is very useful for these patients as it can facilitate treatment at home; consequently, they are able to avoid visiting community centers and hospitals. Thus, virtual reality has a major role in COVID-19 mitigation, especially for extremely vulnerable people [18]. The different applications of VR are provided in Table 2. 

#### 3.3.5. Tele-Pharmacy Services 

Digitalization of prescribing or refilling medications was already in place even before the pandemic, but the existence of the pandemic has helped to improve and broaden its applications. Also, tele-pharmacy services play an important and effective role in controlling the pandemic by increasing access to care. Remote prescription is considered as a measure to prevent infection in extremely vulnerable patients, such as cancer patients [56]. The cross-sectional data from the literature showed that remote prescription has an 88% satisfaction and acceptance rate in those patients [56]. Another prospective study for four months to assess predictors of effective remote prescriptions services included 52 community pharmacies. The results showed that remote prescription services are an effective tool to minimize the pressure on the healthcare system during the pandemic [14]. This application of digital health might be the mainstream mode of prescribing or refilling medications for chronically ill patients. 

#### 3.3.6. Artificial Intelligence Helps in Decision Making 

Artificial intelligence (AI) is also considered one of the innovative new forms of technology that has helped to fight the pandemic [57]. It has enabled fast and reliable screening and diagnosis of people with symptoms such as fever and cough. A group of researchers evaluated the ability of AI to detect cough sounds in COVID-19 patients, and the results showed that AI can distinguish between COVID and non-COVID coughs in two seconds [12]. AI, therefore, can provide rapid information to make a medical decision.

Also, AI has helped to decrease the workload of healthcare practitioners during the pandemic [58]. Research has found that using AI has offered helpful solutions for administrative and clinical practice, such as the triage of patients, minimizing physical contact with COVID-19 cases, digitalization of clinical investigations, and assisting in the training of practitioners [59]. AI is also effective in contact tracing, localized quarantining and self-isolating, and safety notifications [59]. This new paradigm of patient diagnosis and patient treatment could lead the future of medical treatments. 

#### 3.3.7. Machine Learning Has Helped in Predictions, Expectations, and Planning

Machine learning (ML) and mathematical modeling are under the umbrella of AI [60,61] and are mostly used in epidemiological sciences [60]. ML, for example, has been used extensively in predicting and anticipating the future of the pandemic based on the available data and data derived from the literature, or real-time data [61]. ML has been used to manage the pandemic by providing expectations and estimations of how the pandemic is impacting the healthcare systems, mortality and infection rates, how many waves are expected, and when it will be possible to ease public restrictions and lockdowns [61]. This valuable information contributes to planning and decision making. An example from the literature was introduced by imperial scientists [32]. Soon after the beginning of the pandemic, scientists expected the second wave in England before it happened [32,62]. This allowed enough time for healthcare systems and authorities to prepare and to take the best actions and measures.

A group of researchers from Saudi Arabia and the United Kingdom used a mathematical model at the beginning of the pandemic (to evaluate the effectiveness of public restrictions on daily COVID-19 cases). The model was based on the Exponential Logistic Growth and Susceptible Infectious Recovered (SIR) model. Researchers concluded that the rates of infection and death would increase for the following months (April and May 2020) despite the public restrictions that were already in place [63,64].

### 3.4. What Are the Facilitators and Barriers That Contribute to the Usability of Digital Technologies during the COVID-19 Pandemic? 

Unlike previous health situations, adaptation and diffusion of healthcare innovations have been growing quickly due to the COVID-19 pandemic. That improvement includes multiple-level users such as organizations, policymakers, industries, healthcare providers, and consumers [13].

#### 3.4.1. Government Support 

It was obvious that there was pressure from healthcare authorities and governments to adopt telehealth ecosystems to facilitate public measures and self-quarantining. This was supported by many initiatives in the healthcare systems, resulting in a shift in the healthcare models from the conventional to the modern within a fleeting period. For example, in the UK, the government initiated a key statement: “*Stay home, protect the NHS, and save lives*” to curb the spread of the virus. Another example from Saudi Arabia is when the government initiated a statement: “*We are all responsible*” to fight the coronavirus [63,65]. This has helped directly and indirectly to accelerate the adoption and diffusion of digital technology in healthcare systems, which would have taken years if the pandemic did not exist.

#### 3.4.2. Technology and Computer Literacy 

Another facilitator is that the pandemic has happened at a time when individuals of all ages have wide access to advanced digital technology. Technology literacy and usability were already at a higher level than in previous years. Producers had designed simpler devices (smartphones and tablets), internet communication was available everywhere, internet and smart devices were affordable, and trust in technology to receive care had increased [27]. All these factors together facilitated the adoption and diffusion of digital health technology to curb the COVID-19 pandemic [27]. However, this might not be the case in the countries where telehealth applications have already been challenging before the COVID-19 outbreak. 

#### 3.4.3. Faster Vaccine Development

Adding to that, new digital technology (mRNA) has enabled faster development of vaccines than conventional methods [66,67]. This has paved the road for pharmaceutical companies to apply novel technology in vaccine developments to produce an effective COVID-19 vaccination [68,69]. More than 3.22 billion vaccines have been administered around the world, which has helped to control the pandemic [67]. 

#### 3.4.4. Access Real-Time Data 

Using the digital health platforms on a large scale during the pandemic has offered numerous opportunities to access real-time data. For example, Iceland, South Korea, and Germany have collected data on asymptomatic COVID-19 patients via mobile apps [20]. These data have been linked to other clinical datasets, for example, genome sequencing data, to provide robust information about the pathology of the virus [70]. Furthermore, real-time data about symptomatic and asymptomatic COVID-19 cases have helped to visualize the prevalence of the virus and to discover new variants [70].

#### 3.4.5. Cost-Effectiveness 

Without a doubt, the pandemic has placed a big economic burden on most countries, and, in some locations, it has destroyed what has already been achieved to develop healthcare systems. However, digital technology has helped to minimize the cost burden in some countries such as Taiwan. A group of researchers from Taiwan evaluated the cost-effectiveness of digital health solutions to monitor Taiwanese travelers for 14 days. The study shows that digital health technology successfully provides timely disease management, with lower costs for travelers who pay for travel insurance [71]. Daily symptoms screening for suspected or positive COVID-19 patients has helped to provide rapid disease management by providing the required support and preventing the consequences of hospital admission [72]. There is evidence that daily screening of symptoms via digital health solutions, in conjunction with a polymerase chain reaction (PCR) test, can reduce the number of new COVID-19 cases. This approach also helps to reduce the cost compared to other strategies such as hospital screening or providing no interventions.

#### 3.4.6. Barriers 

Despite the opportunity for quick implementation of digital health technology, using digital technology in healthcare remains a challenge for healthcare systems in poor countries where the infrastructure for technology is very poor [6].

#### 3.4.7. Healthcare Systems Complexity 

Reviewing the quick adoption of healthcare innovations during the pandemic demonstrates the barriers related to the complexity of healthcare systems, especially in poor countries or countries that have vulnerable healthcare systems. Beliefs and perceptions about digital health were a challenge before, during, and beyond the pandemic. Internet connections, digital health support, data security and privacy, and economic status are also considered barriers [25]. These barriers emerge from multiple-level users, including organizations, healthcare providers, policymakers, and consumers [9]. Nevertheless, attention must be paid to poor healthcare systems in which the users barely receive the basic healthcare that they need. 

#### 3.4.8. Lack of Financial Support

The rapid change during the COVID-19 pandemic from face-to-face to digital health might result in greater financial pressure on healthcare systems by making healthcare delivery costlier than usual because of the greater need for adequate resources and trained healthcare providers. For example, hospitals and healthcare centers will need to be prepared to send individuals to their homes with medical equipment and to monitor them [27]. Furthermore, the medical equipment will need operation, supervision, and maintenance [25]. 

#### 3.4.9. Absent of the Guidelines 

The risks and regulations of adopting digital health have increased due to the trend of the pandemic. Most of the digital health solutions were executed within a short period to cope with self-isolation standards or to support the public measures and lockdowns. This has resulted in the requirement of guidelines to adopt and operate digital solutions in healthcare systems. Some of the systems already have guidelines that are being used in clinics, but others do not [73]. For example, in poor countries such as sub-Saharan countries, there have been scant guidelines for digital health during the pandemic, which has resulted in issues regarding patient privacy [25]. International and local guidelines are needed to adopt digital technologies in clinical settings. This will encourage patients to trust digital health and increase usability [74]. This has also increased the lack of adherence to digital health technologies and increased the stress on the healthcare providers and patients [74].

## 4. Limitations 

We reviewed articles evaluating digital technology with COVID-19 as well as articles with varying study outcomes. Future research may explore digital technologies with COVID-19 within specific clinical outcomes (e.g., cost effectiveness). 

## 5. Conclusions and Future Perspectives

To enhance these digital health applications during the pandemic and beyond, stakeholders must consider the following recommendations. First, sustainability of digital health technology in routine care in the face of future pandemics. Second, analyzing the cumulative experience from the use of digital health technology will enable the provision of accessing healthcare and the control of future pandemics. Third, develop digital health solutions that are more affordable and easier to use. Fourth, consider innovations and digital health technology in training and routine care practice. Fifth, user-led innovations should be included in the design and development of telehealth solutions. Sixth, clinical research should be supported with adequate resources and funds to explore the long-term benefits of using telehealth to reduce the cost of healthcare services. Seventh, patient concerns and needs should be considered when designing and distributing digital health technology. Eighth, patient privacy and confidentiality must be secured during data transformation and telemonitoring. Finally, standard guidelines must lead the implementation of digital health platforms in new healthcare systems.

In conclusion, digital health during the COVID-19 pandemic has been identified as an essential tool to mitigate the transmission of the virus. Digital technology has provided different applications and has played various roles in curbing the pandemic. Stronger healthcare systems have had a more rapid implementation of digital technology during the pandemic than poor and vulnerable healthcare systems. Several facilitators and barriers have been identified in the literature regarding the implementation of digital health during the COVID-19 pandemic.

## Figures and Tables

**Figure 1 ijerph-19-08287-f001:**
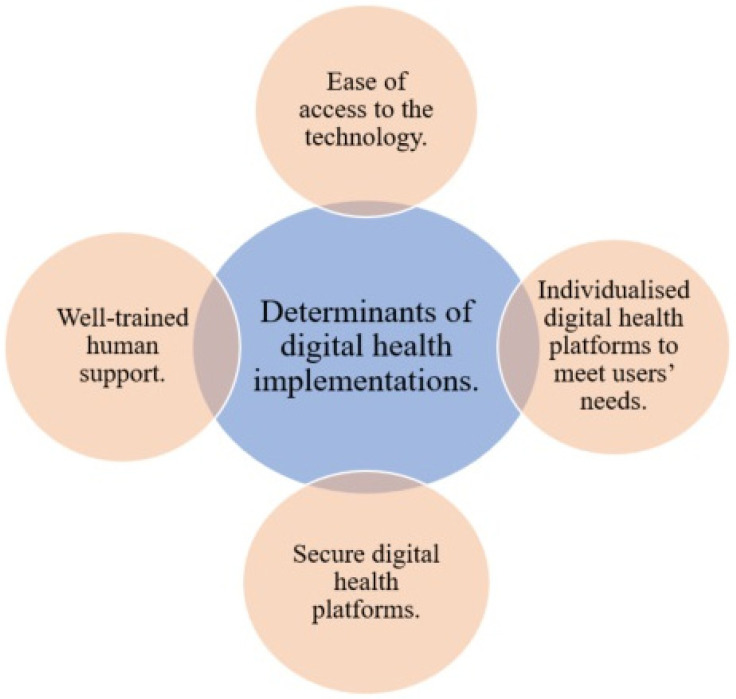
Determinants of digital health.

**Table 1 ijerph-19-08287-t001:** Examples of current mobile health apps to control the spread of COVID-19.

App Name	Objective	Target Providers/Patients	Location
Seha	Daily symptoms, online appointments, and health records	Patients, doctors, and healthcare providers	Saudi Arabia [41,42].
Arogya Setu	Daily symptoms and real-time data	Patients	India [43].
HeadToToe	Education and coronavirus-related information and practice	Healthcare providers	Switzerland [44].
Elisabeth Twee Steden Behandelwijer	Daily symptoms, self-assessment, and coronavirus-related information	Policymakers, healthcare providers, and patients	The Netherlands [45].
NHS COVID-19	Daily symptoms, controlling social distance, test results, and data sharing	Policymakers, healthcare providers, and patients	United Kingdom [46].

Note: NHS: national health system.

**Table 2 ijerph-19-08287-t002:** Different applications of virtual reality.

Field	Distributions/Uses
Doctor training	Support the training of doctors regarding the COVID-19 virus [52].
Pulmonary rehabilitation	Speed up the recovery rate of COVID-19 patients [53].
Physiotherapy	Virtual reality helps physiotherapists to handle cases of rehabilitation therapy more effectively [18].
Managing chronic pain	Support COVID-19 patients with chronic pain to achieve relief [54].
Mental rehabilitation/support	Support people suffering from mental issues, depression, and anxiety [55].
Psychological support	Support patients who are socially isolated, or who have phobias, fear, and any other psychological illnesses [55].

## Data Availability

Not applicable.

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
