# Peer review of "The Role of Digital Technology in Curbing COVID-19"

_ijerph, 2022, doi:10.3390/ijerph19148287_

Round 1

Reviewer 1 Report

The topic of this study is a review paper that examines the role of Digital Technology in dealing with COVID-19, which is very interesting and academically contributive. However, I think it has the same format as a research paper in constructing the content, not review paper. Please refer to the review papers of this journal and revise it accordingly.

Rather than presenting research questions in the introduction, the necessity and purpose of the paper to examine these contents should be dealt with in more depth.

The presence of table 1 in the introduction only confuses reading the content. Please put it as an appendix.

In Methods section, authors mentioned ”Data analysis was carried out by summarising and synthe-99 sising the current data. Titles, abstracts, and full text were reviewed by two independent 100 reviewers. Full search strategy was provided in the Appendix 1.” However, It is hard to understand the process of the review of this article.

In result section, authors mentioned “There were 57 articles included in our review, which were summarized under the 103 following headings (Figure 1 and Figure 2).” I think that this is related to method section.

The appendix should follow after the references.

Author Response

Reviewer1

The topic of this study is a review paper that examines the role of Digital Technology in dealing with COVID-19, which is very interesting and academically contributive. However, I think it has the same format as a research paper in constructing the content, not review paper. Please refer to the review papers of this journal and revise it accordingly.

  • Thank you for your comment. It has been corrected in both the title and methods of the paper. 
  • Additional references have been added to the introduction.

Rather than presenting research questions in the introduction, the necessity and purpose of the paper to examine these contents should be dealt with in more depth.

  • Thank you for your comment. Our introduction has been revised. We hope the clarification is clearer now.

The presence of table 1 in the introduction only confuses reading the content. Please put it as an appendix.

  • Thank you for the valuable comment. The online supplementary now contains this information.

In Methods section, authors mentioned ”Data analysis was carried out by summarising and synthe-99 sising the current data. Titles, abstracts, and full text were reviewed by two independent 100 reviewers. Full search strategy was provided in the Appendix 1.” However, It is hard to understand the process of the review of this article.

  • Thank you for the comment. Our revision clarifies the methods. We Hope it is now clearer and more precise.

In result section, authors mentioned “There were 57 articles included in our review, which were summarized under the 103 following headings (Figure 1 and Figure 2).” I think that this is related to method section.

  • Thank you for the comment. The results section has been updated to reflect this. We improved the clarity and precision of the results section.

The appendix should follow after the references.

  • Thank you for the suggestion. The revised manuscript has no appendices.

Reviewer 2 Report

An increasing number and variety of digital health applications have been available throughout the pandemic, such as telehealth, smartphone mobile health apps, machine learning, and artificial intelligence. Each technology has played a particular role in curbing COVID-19 transmission.

This referee found this article interesting and a vast list of articles were reviewed. Nevertheless, formal aspects were not respected and they need to be solved before publication. The following are some suggestion to focus the readability of the article.

The authors should clear clarify the limitations of their study and the mening for “future research needs to explore digital technologies with COVID-19 within specific study designs (e.g., clinical trials, cross-sectional studies) and report the results based on quality assessment and standard guidelines”.  How authors are expecting to access the “quality of the studies" in future studies.

Figures legends present information regarding the source and copyright. These data are not needed.

Moreover Appendix about search strategy is not relevant in this article since its content is rapidly outdated.

The most problematic topic is the Results Section. Indeed, the authors are discussing the results with several citations and not presenting a formal results section.

This is not a systematic review. Therefore, a results section just to say that “57 articles were included in this review” is not appropriate. Independent sections of the topic 3.1 and subsequent ones are better and more appropriate than considering those topicsunder “results section”.

The style for references citation should follow journal instruction for authors.

Finally, conclusion section should focus major raised issues, perhaps by fusing “recommendations” with “conclusions” in a single “Conclusions and future perspectives” section

a. Which digital technologies have been used during the COVID-19 pandemic to mitigate the transmission of the virus?

b. To what extent have digital technology applications helped to mitigate the spread of COVID-19?

c. Which facilitators and barriers have contributed to the usability of digital technologies during the COVID-19 pandemic?

Author Response

Reviewer 2

An increasing number and variety of digital health applications have been available throughout the pandemic, such as telehealth, smartphone mobile health apps, machine learning, and artificial intelligence. Each technology has played a particular role in curbing COVID-19 transmission.

This referee found this article interesting and a vast list of articles were reviewed. Nevertheless, formal aspects were not respected and they need to be solved before publication. The following are some suggestion to focus the readability of the article.

  • Thank you for the you for your valuable input. It helps to strengthen the presentation and clarity of the manuscript.

The authors should clear clarify the limitations of their study and the mening for “future research needs to explore digital technologies with COVID-19 within specific study designs (e.g., clinical trials, cross-sectional studies) and report the results based on quality assessment and standard guidelines”.  How authors are expecting to access the “quality of the studies" in future studies.

  • Thank you for your comment. We revised the limitation section as suggested. We hope it is clear now.

Figures legends present information regarding the source and copyright. These data are not needed.

  • Thank you for the comment. This information was removed from the figures.

Moreover Appendix about search strategy is not relevant in this article since its content is rapidly outdated.

  • Thank you for the comment. This has been removed from the online supplement.

The most problematic topic is the Results Section. Indeed, the authors are discussing the results with several citations and not presenting a formal results section.

  • Thank you for the comment. We accept that we have not done formal presentation by clinical outcome (i.e. hospitalisation), because we thought this is the best headings to summarise the findings. We summarise the findings under each objective separately to make it easy to follow, given our argument on the validity of the presentation we have reported, both authors have an agreement that these headings in the result section were the most appropriate to report results. 

This is not a systematic review. Therefore, a results section just to say that “57 articles were included in this review” is not appropriate. Independent sections of the topic 3.1 and subsequent ones are better and more appropriate than considering those topicsunder “results section”.

  • Thank you for your comment. In the revised manuscript, this has been corrected. The result section has been updated to be clearer and more precise..

The style for references citation should follow journal instruction for authors.

  • Thank you for your comment. The reference style has been corrected.

Finally, conclusion section should focus major raised issues, perhaps by fusing “recommendations” with “conclusions” in a single “Conclusions and future perspectives” section

a. Which digital technologies have been used during the COVID-19 pandemic to mitigate the transmission of the virus?

b. To what extent have digital technology applications helped to mitigate the spread of COVID-19?

c. Which facilitators and barriers have contributed to the usability of digital technologies during the COVID-19 pandemic?

- Thank you for the comment. We have incorporated this section into one single section as “conclusion and future perspectives”

Thank you

Round 2

Reviewer 1 Report

Thank you for submitting your revised manuscript. 

However, I do not think that your manuscript is improved in methods section.

Still, I can not understand how you processed this research.

Author Response

  • Thank you for your comment. We have added a sentence to the methods which clarify the processing of the data. We stated, “The final findings were reported in narrative form.” Hope it is clear now.

Reviewer 2 Report

Additional comments are not needed. Authors carefully answered raised comments.

Author Response

  • Thank you. We appreciated taking the time to review our manuscript and helping us improve it.